# Reuse of Wasted Bread as Soil Amendment: Bioprocessing, Effects on Alkaline Soil and Escarole (*Cichorium endivia*) Production

**DOI:** 10.3390/foods11020189

**Published:** 2022-01-11

**Authors:** Claudio Cacace, Carlo Giuseppe Rizzello, Gennaro Brunetti, Michela Verni, Claudio Cocozza

**Affiliations:** 1Department of Soil, Plant, and Food Science, University of Bari, 70126 Bari, Italy; claudio.cacace@uniba.it (C.C.); gennaro.brunetti@uniba.it (G.B.); claudio.cocozza@uniba.it (C.C.); 2Department of Environmental Biology, “Sapienza” University of Rome, 00185 Rome, Italy; carlogiuseppe.rizzello@uniroma1.it

**Keywords:** wasted bread, bioprocessing, lactic acid bacteria, soil amendment

## Abstract

In an era characterized by land degradation, climate change, and a growing population, ensuring high-yield productions with limited resources is of utmost importance. In this context, the use of novel soil amendments and the exploitation of plant growth-promoting microorganisms potential are considered promising tools for developing a more sustainable primary production. This study aimed at investigating the potential of bread, which represents a large portion of the global food waste, to be used as an organic soil amendment. A bioprocessed wasted bread, obtained by an enzymatic treatment coupled with fermentation, together with unprocessed wasted bread were used as amendments in a pot trial. An integrated analytical plan aimed at assessing (i) the modification of the physicochemical properties of a typical Mediterranean alkaline agricultural soil, and (ii) the plant growth-promoting effect on escarole (*Cichorium endivia* var. *Cuartana*), used as indicator crop, was carried out. Compared to the unamended soils, the use of biomasses raised the soil organic carbon content (up to 37%) and total nitrogen content (up to 40%). Moreover, the lower pH and the higher organic acid content, especially in bioprocessed wasted bread, determined a major availability of Mn, Fe, and Cu in amended soils. The escaroles from pots amended with raw and bioprocessed bread had a number of leaves, 1.7- and 1.4-fold higher than plants cultivated on unamended pots, respectively, showing no apparent phytotoxicity and thus confirming the possible re-utilization of such residual biomasses as agriculture amendments.

## 1. Introduction

Many agricultural soils are characterized by a low content of organic matter (OM) that represents a limiting factor for crop growth and production [1]; moreover, the OM decomposition rate also increases with a warm climate and intensity of cultivation [2]. In the perspective of sustainable agriculture, the reuse of organic waste as soil amendments is a promising tool to recover soil fertility [3]. Several soil properties, such as pH, nutrient availability, structure and water infiltration, long-term carbon sequestration, and soil biological activity, are positively influenced after the addition of organic amendments [4].

A wide range of food waste and by-products with high content of OM represent an underutilized resource for agronomic applications [5], i.e., among some Mediterranean countries, bread and bakery products represent up to 20% of the total daily food waste produced by some surveyed consumers [6]. Melikoglu and Webb [7] estimated that the bread wasted daily worldwide is around hundreds of tons, and only a little quantity is reused mainly to feed livestock. The loss of bread occurs through the entire supply chain, not only at the household level: for example, during sandwich production, crusts and external layers removed from loaves represent up to 40% of the products [8]. Recently, the possibility of reusing wasted bread (WB) as a substrate for the cultivation of lactic acid bacteria (LAB) to be used as starters for the food industry was investigated [8]. Since LAB causes fast acidification through the production of organic acids, an acidified biomass, employed as a soil amendment, could be of interest for alkaline soils, such as Mediterranean ones. In such pH conditions, many essential plant nutrients are not available for crops, e.g., phosphorous precipitates as Ca phosphates [9], but the competition for the sorption sites between P and organic acids helps to increase P availability [10]. The same occurs for another important nutrient, iron, that in alkaline and oxygenated soils precipitates as iron oxides [11].

Recent scientific evidence confirmed LAB as plant growth-promoting microorganisms (PGPM); besides indirectly improving nutrient acquisition, they can act as biocontrol agents, improving the ability of the host plant to withstand biotic and abiotic stress, or by producing compounds that directly stimulate plant growth [12]. As for most PGPM, plant growth promotion is the simultaneous result of multiple biochemical mechanisms [12]. In addition, former *Lactobacillus* spp. is among the bacterial species able to bioaccumulate metals. Maintaining crop production within a context of land degradation, changing climate, and a growing population is of utmost significance. A rapid and more efficient transformation of the agricultural system that guarantees high-yield production with continually limited resources is therefore required.

In this framework, this study aimed at investigating the potential of wasted bread to be used as an organic soil amendment. A bioprocessed wasted bread (bWB), obtained by an enzymatic treatment coupled with fermentation and containing viable LAB cells at high cell density, together with a biomass of unprocessed wasted bread, were included in this study and used in a pot trial. As an effect of either the organic soil amendment or the viable microorganism supplementation, the physicochemical properties of a typical Mediterranean alkaline agricultural soil could be modified by bioprocessed wasted bread supplementation; additionally, a growth-promoting effect on escarole, used as an indicator crop, could be achieved. To confirm the above-mentioned hypothesis, an integrated analytical approach aimed at assessing the main characteristics of both soil and plants was carried out.

## 2. Materials and Methods

### 2.1. Bread-Based Amendments Preparation and Characterization

#### 2.1.1. Raw Material, Enzymes, and Microorganisms

Wasted white wheat bread (surplus from the production) was kindly provided by the industrial bakery Vallefiorita Srl (Ostuni, Italy). Bread, having the following composition: moisture, 12.08%; proteins, 12.89%; fats, 1.55%; carbohydrates, 70.82%; and dietary fibers, 5.21%, was ground into small crumbs (<1 mm), mixed with distilled water (65%), and homogenized to obtain the WB.

Veron^®^ Mac, a maltogenic amylase used in the bakery industry and purchased from AB Enzymes (Darmstadt, Germany), was added to the bread homogenate for bioprocessing. *Lactiplantibacillus plantarum* H64, belonging to the Culture Collection of the Department of Soil, Plant and Food Sciences (University of Bari, Italy), was used as a starter for biomass fermentation. The strain was routinely propagated on De Man, Rogosa, and Sharpe (MRS) (Oxoid, Basingstoke, Hampshire, UK) at 30 °C. When used for fermentation, cells grown until the late exponential phase of growth (circa 12h) were harvested by centrifugation at 9000× *g* for 10 min at 4 °C, washed twice in a sterile physiological solution (NaCl 0.9%, *w*/*v*), and resuspended in distilled water.

#### 2.1.2. Bioprocessing

Bioprocessed wasted bread (bWB) was prepared by mixing ground bread (35%), distilled water (65%), amylase Veron^®^ Mac at the concentration recommended by the manufacturer (3 mg/100 g), and the selected LAB strain at the final cell density of ca. 7 log cfu/g. The biomass was then incubated at 30 °C for 24 h, characterized, and used in pot trials. WB was also characterized and used as a control in all the experiments.

#### 2.1.3. Characterization of Wasted Bread Biomasses

The pH of the biomasses was determined by a pH meter (Model 507, Crison, Milan, Italy) with a food penetration probe, and total titratable acidity (TTA) was determined according to the AACC method 02–31.01 [13] and expressed as the amount (mL) of 0.1 M NaOH necessary to reach pH of 8.4.

Presumptive LAB were enumerated, before and after fermentation, using MRS agar medium (Oxoid) supplemented with cycloheximide (0.1 g/L). Plates were incubated at 30 °C for 48 h, under anaerobiosis (AnaeroGen and AnaeroJar, Oxoid). WB and bWB were also characterized for the presence of yeasts, molds, and *Enterobacteriaceae*. Yeasts and molds were cultivated on Yeast Peptone Dextrose Agar medium (Sigma-Merck, Darmstadt, Germany), supplemented with 0.01% chloramphenicol, through pour and spread plate enumeration, respectively, and incubated at 25 °C whereas *Enterobacteriaceae* were determined on Violet Red Bile Glucose Agar (Oxoid) at 37 °C for 24 h.

Water/salt-soluble extracts (WSE) from wasted bread biomasses were prepared according to the method originally described by Osborne and modified by Weiss et al. [14] using 50 mM Tris–HCl (pH 8.8). After centrifugation, the supernatants were used to determine sugars, organic acids, peptides, and total free amino acid (TFAA) concentration.

Glucose and maltose were measured using the D-Fructose D-Glucose Assay Kit K-FRUGL and the Maltose-Sucrose-D-Glucose Assay Kit K-MASUG (Megazyme International Ireland Limited, Bray, Ireland), respectively, following the manufacturer’s instructions.

Organic acids were quantified by High Performance Liquid Chromatography (HPLC), using an ÄKTA Purifier system (GE Healthcare, Buckinghamshire, UK) equipped with an Aminex HPX-87H column (ion exclusion, Biorad, Richmond, CA), as described by Rizzello et al. [15].

For the analysis of peptides, WSE were treated with trifluoroacetic acid (0.05% *wt*/*vol*), centrifuged (10.000× *g* for 10 min), and subjected to dialysis (cut-off 500 Da) to remove proteins and free amino acids, respectively. Then, peptide concentration was determined by the *o*-phtaldialdehyde method as described by Church et al. [16], and dialysates were analyzed through Reversed-Phase Fast Performance Liquid Chromatography (RP-FPLC) using a Resource RPC column and ÄKTA FPLC equipment with the UV detector operating at 214 nm (GE Healthcare Bio-Sciences AB, Uppsala, Sweden) as described by Rizzello et al. [15]. TFAA was analyzed by a Biochrom 30^+^ series Automatic Amino Acid Analyzer (Biochrom Ltd., Cambridge Science Park, United Kingdom), equipped with a Li-cation-exchange column (4.6 × 200 mm internal diameter) [17].

WB and bWB were analyzed for moisture, ash content, pH, and electrical conductivity (EC) according to the methods previously proposed by Trinchera et al. [18]. In detail, the moisture, expressed as a percentage of the initial weight, was determined by drying samples at 105 °C overnight; the ash content, expressed as a percentage of the dry matter, was determined by combustion in a Controls 10-D1418/A muffle furnace at 550 °C for 12 h. The EC was measured using a Hanna Edge^®^ EC instrument on sample/water extracts (1:10 *w*/*v*) after shaking for 30 min. Total N (TN) content was determined by the Kjeldahl method, while the organic carbon (OC) content was determined by dichromate oxidation and subsequent titration with ferrous sulfate according to Ciavatta et al. [19]. This method is suitable for samples characterized by high OC levels. The total P content was measured spectrophotometrically at 650 nm after incinerating biomass samples at 550 °C, suspending ashes in 10% hydrochloric acid solution, and developing the blue color in the filtered solution according to the Olsen method [20].

### 2.2. Pot Trial

#### 2.2.1. Experimental Design

An alkaline soil, classified Calcic Luvisol according to IUSS Working Group WRB [21], was collected from a stone fruit orchard, air-dried, and used for the pot experiment. The particle size composition of the soil used for the pot trial was 173 ± 2, 356 ± 3, and 471 ± 3 g kg^−1^ of sand, silt, and clay, respectively, corresponding to clayey texture according to the Soil Survey Staff methodology [22].

For the pot trial, two experiments were carried out, one with plants and one without, aiming at assessing whether the changes observed in the soil features were a consequence of the amendment addition or the soil ecosystem interaction with the plant (rhizosphere effect). Treatments included in the pot experiment were: (i) not amended soil, without a plant (CTA); (ii) soil amended with WB, without a plant (WBA); (iii) soil amended with bWB, without a plant (bWBA); (iv) not amended soil, with a plant (CTP); (v) soil amended with WB, with a plant (WBP); (vi) soil amended with bWB and with a plant (bWBP). Pots (0.4 L each) were distributed in a completely randomized design with three replications for each treatment, for a total of 18 experimental pots. The trial was performed in a cold greenhouse at the University of Bari (South Italy). The amended pots received WB or bWB at a dose of about 25,000 kg ha^−1^, according to the good local agricultural practices [3].

Thirty-day-old seedlings of *Cichorium endivia* var. *Cuartana*, a variety of escarole, were transplanted at the end of the first period of February 2020 and the trial was stopped at the beginning of April. The first irrigation was performed immediately after the transplanting for the rooting and establishment of the plants. During the trial, the temperature ranged from 5 °C at night to 23 °C at mid-day.

#### 2.2.2. Soil Characterization

The soil was characterized at the beginning of the trial (T0) by means of pH, EC, TN, available phosphorous (P_ava_), Mn, Fe and Cu, and OC content. The pH was measured in deionized water (pH_H2O_) and in 1 M KCl (pH_KCl_) suspensions at 1:2.5 soil to liquid ratio, whereas the electrical conductivity (EC) was measured in filtrates from a 1:2 soil to water ratio. The TN content was determined by the Kjeldahl method. The OC was measured by dichromate oxidation and ferrous sulfate titration according to the Walkley-Black method [23]. The P_ava_ was extracted with a 0.5 M NaHCO_3_ solution and determined spectrophotometrically at 650 nm [20]. Diethylenetriaminepentaacetic acid (DTPA)-extractable fractions of Mn, Fe, and Cu were obtained from a 1:2 soil to DTPA solution. DTPA extracts were filtered by gravity through Whatman No. 42 filter paper, and the solutions were then analysed using an inductively coupled plasma iCAP 6000 Series ICP-OES Spectrometer (Thermo Electron Corporation, Walthman (MA), USA). The soil texture was identified using the Soil Survey Staff methodology [22].

At the end of the experiment, all soils were characterized again to investigate the effects of WB, bWB, and/or plants on the soil parameters with respect to T0.

#### 2.2.3. Plant Characterization

To verify the effects of WB and bWB on the soil/plant system during the test, indirect measurements of the chlorophyll content were carried out using SPAD-502 (Konica Minolta, Japan). At the end of the test (50 days from transplanting), the total number of plant leaves and the fresh weight of the plant were determined to calculate the yield of each treatment. Moreover, leaf samples were analyzed for their P, Mn, Fe, and Cu content, aiming at verifying the effects of each treatment on leaf composition, the total P was obtained as described above for the wasted biomasses. The total Mn, Fe, and Cu content were determined using the microwave-assisted acid digestion method, adding a Suprapur^®^ HNO_3_:H_2_O_2_:HCl mixture (6:1:1, *v*:*v*:*v*) to each sample. At the end of the digestion, the samples were cooled, filtered through Whatman No. 42 filter paper, and diluted with distilled Milli-Q Reagent grade water and, finally analyzed by means of an inductively coupled plasma iCAP 6000 Series ICP-OES Spectrometer (Thermo Electron Corporation).

### 2.3. Statistical Analysis

Experimental data were tested against the normal distribution of variables (Shapiro—Wilk test) and the homogeneity of variance (Bartlett test) using R studio. The variables normally distributed with homogeneity of variances verified were subjected to an ANOVA and HSD test. Data not normally distributed were subjected to the Levene test and a no-parametric ANOVA analysis (Kruskal-Wallis test) and Dunn test.

## 3. Results

### 3.1. Amendment Characterization

In bWB, the initial cell density of presumptive LAB corresponded to the targeted inoculum and increased after 24 h of fermentation approximately 2 log cycles, reaching 9.35 ± 0.36 log cfu g^−1^.

The amendments were also characterized for the presence of yeasts, molds, and *Enterobacteriaceae*. Yeasts and molds in WB were 3.8 ± 0.1 and 2.9 ± 0.0 log cfu g^−1^, respectively, whereas *Enterobacteriaceae* were 4.1 ± 0.2 log cfu g^−1^. After bioprocessing, all microbial species investigated were in a notably lower range, compared to WB. Yeasts and molds remained below 2 log cfu g^−1^, whereas *Enterobacteriaceae* were not detected.

Relevant acidification was obtained after fermentation. The pH decreased from 5.92 ± 0.09 of WB to 3.89 ± 0.16 of bWB, with a production of 56 and 2 mmol kg^−1^ of lactic and acetic acid, respectively, which were detected in traces in unprocessed WB. As expected, the TTA value was significantly higher in bWB (8.46 ± 0.48 mL) compared to WB (1.54 ± 0.08 mL). Glucose and maltose, present in small amounts in WB (0.95 and 2.79 mg g^−1^, respectively), were also found at higher concentrations in bWB (12.87 and 7.83 mg g^−1^, respectively).

Peptide content in WB was 101.36 ± 4.29 mg g^−1^ and increased by ca. 38% during fermentation. This trend was confirmed by the FPLC chromatograms, where although no differences among peak area/total area ratios were observed at different percentages of acetonitrile, the total area and the number of detected peaks increased by roughly 20% in bWB compared to WB. On the contrary, TFAA slightly but significantly (*p* < 0.05) decreased after fermentation (ca. 700 mg g^−1^). Nevertheless, γ-aminobutyric acid (GABA) content was higher in bWB (149.68 ± 1.36 mg g^−1^) compared to WB (116.39 ± 3.12 mg g^−1^).

WB bioprocessing slightly increased the OC content (7%) compared to unprocessed WB (Table 1). In contrast, WB biomass showed significantly higher EC and TN content than bWB, up to 6% and 9%, respectively. Accordingly, the C/N ratio was significantly higher in bWB than WB. Finally, even though bWB resulted in a numerically higher total P content (2150 ± 15 mg kg^−1^) compared to WB (1716 ± 246 mg kg^−1^), they were not significantly different.

### 3.2. Soil Characterization

The physicochemical properties of cultivated and uncultivated soils treated with the biomasses in comparison to CTP and the soil at the beginning of the trial (T0) are reported in Table 2.

The pH_H2O_ of T0 and CTP was alkaline and ranged from 8.07 ± 0.05 to 8.20 ± 0.15, while bWB and WB supplementation significantly reduced the pH_H2O_ by roughly 8%, even if they did not show significant differences between each other. No significant differences were observed for the pH_KCl_ among all treatments as well (Table 2).

The EC value of cultivated pots of escarole (CTP) significantly increased (417 ± 103 μS cm^−1^) compared to T0, and was further enhanced by the addition of the two biomasses. However, the two amended soils did not show significant differences between each other.

The soil was positively and significantly influenced by the amendments since the treated soils had higher OC and TN content, reaching up to 23% higher values at the end of the trial compared to the soil at T0, whereas CTP showed the lowest TN and OC content (Table 2). The P_ava_, on the other hand, was not significantly influenced by the amendments.

As observed for the cultivated soils, the absence of the plants resulted in very similar trends of pH, EC, OC, and TN, meaning that those parameters were influenced mainly by the biomasses. The availability of P was significantly and negatively influenced by the treatments since, compared to CTA and the soil at the beginning of the trial, a decrease of up to 23% and 20%, respectively, was observed in uncultivated pots (Table 2). Nevertheless, P_ava_ content did not show any statistical difference among samples in cultivated soils.

The bioavailability of Mn, Fe, and Cu in soils with and without plants was also studied (Table 3). In the amended but not cultivated soils, biomass supplementation led to an increase in the availability of Mn and Fe, which were almost 3- and 2-fold higher compared to CTA, while no significant differences were observed for the available Cu among treatments.

The cultivated pots showed the higher availability of the selected elements, up to 36%, 34%, and 13% higher concentrations for Mn, Fe, and Cu, respectively, with bWBP showing the highest values between the amendments.

### 3.3. Plant Characterization

Table 4 reports the mean biometric features of plants at the end of the experiment. The escaroles from WB and bWB amended pots had a number of leaves 1.7- and 1.4- times higher, respectively, than plants cultivated on CTP pots. In particular, WBP and bWBP pots had a yield of 1.95 and 1.70 times higher than CTP, respectively.

The indirect measurement of the chlorophyll content (Spad units) confirmed the biometric results (Figure 1). During the first 25 days after transplantation (DAT), all treatments did not show significant differences in the spad values even if an increasing trend could be observed for the bWB amended pots already from 22 DAT. From 27 to 29 DAT, bWB pots showed the highest spad values, followed by WB, while CTP escaroles had the lowest chlorophyll content. From 32 DAT until the end of the trial, plants amended with both biomasses resulted in the highest spad values, while CTP showed a slow decline of the chlorophyll content.

Lastly, the P, B, Mn, Fe, and Cu content of escarole leaves at the end of the trial, was also measured (Table 5). As observed for the available P of the corresponding soils, the application of WB and bWB significantly decreased the P concentration in escarole leaves by 63% and 59%, respectively, compared to those collected from the CTP pots. Similarly, the B content was 2 and 7 times higher in leaves from CTP compared to WBP and bWBP, respectively. Mn, Fe, and Cu concentration did not differ significantly among leaves of all treatments.

## 4. Discussion

The long-term sustainability of food chains and the management of high levels of food loss and waste are among the challenges the global agri-food system have been facing in recent decades [24]. Bread, whose predicted production volume in 2021 was 209,874.8 million kilograms [25], represents a large portion of the global food waste, with economic and environmental repercussions [26]. The valorization of bakery waste as a food ingredient has been largely investigated recently, and different innovative biotechnological protocols have been proposed aiming at obtaining glucose syrup [27,28] or beer [26,29]. Recently, bioprocessing, e.g., enzymatic treatments and microbial fermentation, have been used to convert bread waste into valuable food ingredients, aiming at the improvement of the technological and sensory characteristics of the biomass, but also to the in situ enrichment of functional compounds such as dextran (with a positive impact on food texture) [30], antimicrobial compounds [31], and GABA [32].

Bioprocessed wasted bread, thanks to its suitability to be converted into a substrate for the growth of several microorganisms, was successfully used for the production of a medium for food industry starter cultivation [8]. Nevertheless, a major part of the wasted bread is no longer edible, not eligible for human consumption, and therefore disposed of as waste, thus representing an environmental issue due to the very high organic load. Only a small part of the wasted bread is employed for ethanol production or re-used as feed [7,26].

In this work, the potential of wasted bread to be used as a soil amendment was investigated. In addition to untreated wasted bread, bread biomass pretreated with enzymes and fermented with selected lactic acid bacteria was considered.

Overall, valorisation of food waste by conversion into products such as biofertilizers and biochar that can be added to the soil for increased nutrient inputs and fertility is gaining attention by the scientific community and industry [33]. Food waste fertilizers can be a relatively cheap source of nutrients compared to commercial inorganic fertilizer sources due to their large availability and the possibility of mass-scale and low-cost production. In addition to the nutrient role of the organic biomasses in soil, food waste can sometimes act as soil amendments, since they are able to affect the PGPM growth and survival, reduce pathogens, release nutrients, reduce leaching, increase water retention, and improve soil structure [34,35]. It was already observed that, as soil amendments, food waste-derived biomasses can increase plant yield and soil productivity [36].

The role of PGPM in soil fertility appears to be crucial; nevertheless, past research only focused on a few groups of common symbiotic rhizosphere microorganisms, such as rhizobia, *Bacillus*, *Pseudomonas*, and mycorrhizal fungi [37]. The functional role of other groups of potential PGPM, including LAB, has not been largely investigated [12], although such microorganisms could represent a genetic and metabolic resource for the development of biochemical solutions to pressing agricultural issues [12]. LAB are ubiquitous members of many plants, soil, and compost microbiomes, but little is known about the functional interactions between the LAB and their hosts.

LAB were shown to solubilize phosphate [38,39], likely through the production of organic acids, and it was also hypothesized that they can fix atmospheric nitrogen [39] or produce siderophores [38]. LAB could act as biocontrol agents; through the production of antimicrobial compounds, reactive oxygen species, and bacteriocins; by excluding pathogens by pre-emptively colonizing plant tissues vulnerable to infection and by altering the plant immune response [12]. Among LAB features, of interest not only from an agronomic but also from environmental purposes, the ability of *Lactiplantibacillus plantarum* to absorb Ni^2+^ and Cr^2+^ (from industrial wastewater) on the surface and inside their cells was proposed by Ameen et al. [40]. The superficial adsorption is possibly due to the electrostatic interaction of metals with the functional groups of the bacterial cell wall [41].

In this framework, the inoculum of wasted biomasses with properly selected LAB could guarantee the dominance of the LAB compared to other microbial groups, thanks to their capability to rapidly acidify the substrate and to produce antimicrobial compounds. In particular, the bioprocessed wasted bread harboured a very high population of the starter *L. plantarum* H64. The strain, previously selected for the ability to biosynthesize GABA in a matrix composed of wasted bread and wheat bran, allowed the repurpose of two of the main by-products of the cereal industry, promoting their application as a bread ingredient [32]. As expected, because of the starter carbohydrate metabolism, the bioprocessing enabled the production of organic acids which are in line with those previously reported in fermented surplus bread matrices [31,32]. On the contrary, proteolysis was not as pronounced if compared to common flour, which is explained by the fact that the proteases of the original flour, composing the bread dough, are degraded during the baking process. Indeed, unlike dairy LAB, most sourdough lactobacilli do not possess a cell-envelope-associated proteinase and depend on cereal-associated proteinase [42]. Hence, in surplus bread matrices, to ease the release of peptides available during LAB fermentation for their catabolism to amino acids or small bioactive sequences, the use of proteases should be considered, as previously reported [31,32]. Although a decrease in the total free amino acid content was observed in bioprocessed wasted bread, GABA content was 28% higher in bWB compared to WB. Additionally, one of the main advantages of the use of fermentation is the ability to prevent the proliferation of other microorganisms, either bacteria or molds, potentially spoiling bread. Indeed, a significant reduction of yeasts, mold, and *Enterobacteriaceae*, was observed after bioprocessing. This is an aspect particularly appealing in terms of the industrial application of bioprocessed biomass since it can guarantee a longer shelf-life of the amendment.

To better understand the principal changes occurring during cultivation, the biomasses were characterized for their main physicochemical properties. The relatively high EC values observed for WB and bWB are probably related to the presence of sodium chloride, commonly added to the bread formulation at 1–2% (*w*/*w*), however, bioprocessing slightly but significantly reduced the EC value of WB, most likely a consequence of sodium lactate formation in presence of a high concentration of lactic acid produced by LAB metabolism. Wasted bread bioprocessing also led to a slight but significant increase and decrease of OC and TN content, respectively. As a result, fermentation determined a positive balance between the C fixed in microbial biomass and the C lost in heterolactic fermentation as CO_2_, whereas a major N loss as NH_3_ through LAB catabolic pathways involving free amino acids could be responsible for the lower TN content in bWB compared to WB [43].

When the biomasses were used as amendments, the soil pH_H2O_ decreased because of the organic acids brought especially by bWB. The higher EC value of the biomasses reflected on that of treated soils, in fact, biomass mineralization could potentially release osmotically active compounds that could have contributed to the EC increase.

The use of biomasses raised the soil OC and TN content compared to the unamended soils. Among cultivated soils, CTP showed the lowest TN content at the end of the trial because of the uptake of nitrogen from the crop against no input. Since the physicochemical parameters (pH, EC, OC, and TN) showed the same trend in cultivated and uncultivated pots, it is safe to assume the biomasses, rather than the plants, were responsible for such changes. On the contrary, the availability of P was influenced by the crop since the absence of plants in amended pots resulted in a reduction of the P_ava_ content compared to all other treatments. The possible explanation for such behaviour is that the application of biomasses enhanced the microbial activity resulting in the immobilization of phosphate as microbial biomass and phytate, the dominant organic P form in soils, that accumulates due to the deficiency of hydrolytic enzymes and precipitates with metal ions [44]. In contrast, the presence of the plants produced a rhizosphere effect which provided phosphatases, responsible for the solubilization of organic P, and suitable organic acids that compete with phosphates for the sorption sites [10]. Indeed, among organic acids the most efficient in solubilizing soil P are the di- and tricarboxylic ones, mainly oxalic and citric acid, while *L. plantarum* H64 employed in the present study mainly produced monocarboxylic acids, such as lactic, acetic, and γ-aminobutyric acids. The lower pH and the higher organic acid content also led to a major availability of Mn, Fe, and Cu in amended soils. These elements, through the ligand exchange, were solubilized from their precipitated oxides, as a consequence of the biomass’s addition as well as the soil microbial community and rhizosphere activities [9].

To evaluate whether soil improvements transmuted to beneficial changes in the plants, escarole growth and composition were monitored. Biometric parameters of escarole plants indicated that WB and bWB promoted plant growth. Even though during the first 25 DAT no significant differences were observed among treatments, probably because of the rooting and establishment of the plants, the following days revealed a trend. Soil amended with bWB was the first to prompt a better chlorophyll content compared to the other treatments because of its higher TN content and the larger supply of GABA that is correlated to several beneficial effects for plants. Indeed, GABA was found to (i) defend roots against pathogens, (ii) serve as an N reservoir, (iii) cryoprotect the tissues, and (iv) synthesize plant hormones [45]. In contrast, the control plants showed the lowest significant SPAD values from 27 DAT until the end of the trial presumably due to nutrient limitation.

The elemental composition of escarole leaves did not reflect the soil availability of the studied elements. In fact, CTP plant leaves showed a P content more than double that of WB and bWB possibly due to the immobilization of P in organic matter. Regardless, this result did not negatively influence the yield of the crop but rather the nutritional value of escarole. To avoid this inconvenience, transplanting should occur later than soil amendment to allow for better mineralization of the organic P. Regarding Mn, Fe, and Cu, their leaf content did not change among treatments although their soil content was higher in amended pots. It is hypothesized that their soil availability was already satisfied in the CTP pots since they are micronutrients, and/or those elements have been accumulated preferentially in the roots.

## 5. Conclusions

In our study, the application of wasted bread, raw or bioprocessed, resulted in higher escarole yield compared to the unfertilized control without any apparent phytotoxicity, thus confirming the possible re-utilization of such residual biomasses in agriculture as amendments. Although the effects could be transient, it is noteworthy that WB and bWB application resulted in a significantly higher soil OC and lower pH_H2O_ value, a feature that can ameliorate the cultivation of alkaline soils (typical of the Mediterranean area) through beneficial effects on the bioavailability of several nutrients. Nevertheless, such biomasses are not suitable for application in acidic soils (in which the excessive bioavailability of micronutrients and the release of Al from mineral weathering could be favoured).

It was previously reported that LAB promote growth in different crops, even though the underlying mechanisms behind this bio-stimulation remain unclear. The use of wasted bread fermented with *Lactiplantibacillus plantarum* resolved the transplantation stress sooner and further investigation is needed to study the effects of such pre-treatment on the standardization of biomass characteristics and its shelf-life. Additionally, WB can be subjected to contamination during storage making the effects of such biomass on soil quality unpredictable.

Even though further studies are necessary to fully exploit the potential of wasted bread as an amendment, the feasibility of its large-scale production is undeniable. Companies currently producing fertilizers could easily handle the collection of the bread from bakeries and large retailer networks thanks to the abundant and widespread availability of this wasted food product. Although a proper technological transfer is needed, the bioprocess proposed is cost-effective and implementable on an industrial scale. The supply to farmers might follow the current sale and distribution channels. In return, bakeries would not have to assume the disposal costs for managing bread waste (also no longer edible or reusable for feed purposes). It can be assumed that the entity of amendment treatment (approximately corresponding to 250 q/ha) could effectively offer a solution to food waste management and the economic and environmental sustainability of agricultural productions.

## Figures and Tables

**Figure 1 foods-11-00189-f001:**
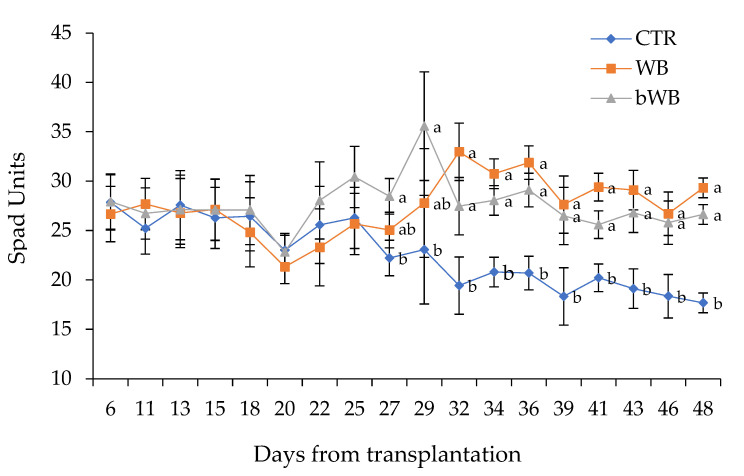
Effect of the biomasses on chlorophyll content of escaroles grown in control soil (CTA), soil amended with wasted bread (WB), and soil amended with bioprocessed wasted bread (bWB, treated with amylase and fermented with *Lactiplantibacillus plantarum* H64). ^a–b^ Different letters indicate significant differences among the data according to the HSD test. Vertical bars represent the standard deviation.

**Table 1 foods-11-00189-t001:** Chemical and physicochemical characteristics of the amendment. WB wasted bread; bWB, bioprocessed wasted bread (treated with amylase and fermented with *Lactiplantibacillus plantarum* H64).

Samples	Moisture (%)	Ash (%)	EC μS cm^−1^	OC (%)	TN (%)	C/N	Total P mg kg^−1^
WB	61.81 ± 6	0.73 ± 0.05	1950 ± 50 ^a^	40.7 ± 1 ^b^	2.47 ± 0.04 ^a^	16.4 ± 0.07 ^b^	1716 ± 246
bWB	65.04 ± 5	0.86 ± 0.06	1820 ± 60 ^b^	43.7 ± 2 ^a^	2.25 ± 0.02 ^b^	19.4 ± 0.03 ^a^	2150 ± 15
	ns	ns	*	*	*	**	ns ^¥^

Data are the means of three independent experiments ± standard deviations (*n* = 3). ^a–b^ Values in the same column followed by a different letter are significantly different according to the HSD test or Dunn test (¥). * Significant at *p* ≤ 0.05; ** Significant at *p* ≤ 0.01, ns: not significant.

**Table 2 foods-11-00189-t002:** Chemical and physicochemical properties of cultivated (P) and uncultivated (A) pots. CT, control soil; WB, soil amended with wasted bread; bWB; soil amended with bioprocessed wasted bread (treated with amylase and fermented with *Lactiplantibacillus plantarum* H64).

Samples	pH_H2O_	pH_KCl_	EC μS cm^−1^	OC g kg^−1^	TN g kg^−1^	P_ava_ mg kg^−1^
*Uncultivated pots*						
**T0**	8.20 ± 0.15 ^a^	7.20 ± 0.08	200 ± 7 ^b^	16.0 ± 0.45 ^b^	1.60 ± 0.10 ^bc^	45.5 ± 1.0 ^ab^
**CTA**	8.20 ± 0.08 ^a^	7.30 ± 0.08	319 ± 57 ^b^	15.2 ±0.51 ^b^	1.5 ± 0.07 ^c^	46.9 ± 1.9 ^a^
**WBA**	7.70 ± 0.07 ^b^	7.30 ± 0.04	805 ± 109 ^a^	20.3 ± 1.47 ^a^	2.1 ± 0.18 ^a^	37.3 ± 2.5 ^bc^
**bWBA**	7.70 ± 0.07 ^b^	7.20 ± 0.02	764 ± 22 ^a^	20.8 ± 0.23 ^a^	1.9 ± 0.12 ^ab^	36.1 ± 2.0 ^c^
	***	ns	***	***	**	**
*Cultivated pots*						
**T0**	8.20 ± 0.15 ^a^	7,20 ± 0.08	200 ± 7 ^c^	16 ± 0.45 ^b^	1.60 ± 0.10 ^bc^	45.5 ± 1.0
**CTP**	8.07 ± 0.05 ^a^	7.25 ± 0.01	417 ± 103 ^b^	17.5 ± 0.49 ^b^	1.31 ± 0.26 ^c^	46.1 ± 2.2
**WBP**	7.67 ± 0.09 ^b^	7.25 ± 0.11	685 ± 109 ^a^	22.4 ± 1.23 ^a^	1.96 ±0.11 ^ab^	48 ± 7.9
**bWBP**	7.57 ± 0.01 ^b^	7.23 ± 0.07	786 ± 56 ^a^	22.4 ± 0.83 ^a^	2.17 ± 0.16 ^a^	41.9 ± 1.8
	***	ns	***	***	**	ns ^¥^

Data are the means of three independent experiments ± standard deviations (*n* = 3). ^a–c^ Values in the same column, among cultivated or uncultivated pots data group, followed by a different letter are significantly different according to HSD test or Dunn test (¥). ** Significant at *p* ≤ 0.01; *** Significant at *p* ≤ 0.001; ns: not significant.

**Table 3 foods-11-00189-t003:** Soil availability of selected micronutrients (mg kg^−1^) in cultivated (P) and uncultivated (A) pots. CT, control soil; WB, soil amended with wasted bread; bWB; soil amended with bioprocessed wasted bread (treated with amylase and fermented with *Lactiplantibacillus plantarum* H64).

Samples	Mn	Fe	Cu
*Uncultivated pots*			
**CTA**	8.06 ± 0.34 ^b^	2.02 ± 0.05 ^b^	1.23 ± 0.01
**WBA**	22.04 ± 6.20 ^a^	2.88 ± 0.48 ^a^	1.37 ± 0.07
**bWBA**	20.88 ± 1.90 ^a^	2.85 ± 0.32 ^ab^	1.49 ± 0.31
	** ^¥^	*	ns
*Cultivated pots*			
**CTP**	10.72 ± 0.85 ^b^	2.08 ± 0.08 ^b^	1.24 ± 0.02 ^b^
**WBP**	16.77 ± 1.65 ^a^	2.87 ± 0.32 ^a^	1.36 ± 0.05 ^ab^
**bWBP**	16.68 ± 2.83 ^a^	3.15 ± 0.21 ^a^	1.43 ± 0.07 ^a^
	*	**	*

Data are the means of three independent experiments ± standard deviations (n = 3). ^a–b^ Values in the same column, among cultivated or uncultivated pots data groups, followed by a different letter, are significantly different according to HSD. test or Dunn test (¥). * Significant at *p* ≤ 0.05; ** Significant at *p* ≤ 0.01, ns: not significant.

**Table 4 foods-11-00189-t004:** Biometric features of escarole plants at the end of the trial.

Samples	Number of Leaves per Plant	Treated/CTP Leaves Ratio	Average Head Escarole Fresh Weight (g)	Treated/CTP Yield Ratio
**CTP**	13 ± 1.15 ^b^	-	6.6 ± 0.47 ^b^	-
**WBP**	22 ± 3.78 ^a^	1.7 ± 0.40	12.9 ± 0.95 ^a^	1.95 ± 0.22
**bWBP**	19 ± 3.05 ^ab^	1.4 ± 0.15	11.2 ± 1.36 ^a^	1.70 ± 0.11
	*	ns	***	ns

Data are the means of three independent experiments ± standard deviations (n = 3). ^a–b^ Values in the same column followed by a different letter are significantly different according to the HSD test. * Significant at *p* ≤ 0.05; *** Significant at *p* ≤ 0.001; ns: not significant.

**Table 5 foods-11-00189-t005:** Micronutrients and phosphorous content expressed as mg kg^−1^, of escarole leaves grown in control soil (CTP), soil amended with wasted bread (WBP), and soil amended with bioprocessed wasted bread (bWBP, treated with amylase and fermented with *Lactiplantibacillus plantarum* H64).

Sample	B	Mn	Fe	Cu	P
CTP	15.45 ± 3.85 ^a^	0.74 ± 0.25	14.77 ± 6.60	0.16 ± 0.03	358 ± 111 ^a^
WBP	7.36 ± 0.96 ^b^	0.76 ± 0.06	15.31 ± 2.93	0.15 ± 0.01	131 ± 50 ^b^
bWBP	2.13 ± 2.52 ^b^	1.06 ± 0.26	10.61 ± 3.24	0.13 ± 0.01	144 ± 29 ^b^
	**	ns	ns	ns	*

Data are the means of three independent experiments ± standard deviations (n = 3). ^a–b^ Values in each column followed by a different letter are significantly different according to HSD.test. * Significant at *p* ≤ 0.05; ** Significant at *p* ≤ 0.01, ns: not significant.

## Data Availability

The data used to support the findings of this study are available from the corresponding author upon request.

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
