# Peer review of "Reuse of Wasted Bread as Soil Amendment: Bioprocessing, Effects on Alkaline Soil and Escarole (Cichorium endivia) Production"

_foods, 2022, doi:10.3390/foods11020189_

Round 1
Reviewer 1 Report
Manuscript ID: foods-1532516
Type of manuscript: Article
Title: Reuse of Wasted Bread as Soil Amendment
Authors: Claudio Cacace, Carlo Giuseppe Rizzello, Gennaro Brunetti, Michela
Verni *, Claudio Cocozza
A brief summary
The article touches on an important topic in its research. The authors approached this problem using well-substantiated methodological experiments. The article is well structured and logical. There are, perhaps, no fundamental remarks, except for some suggestions for revision.
1) This reviewer, being a soil scientist, considers it necessary to indicate a specific soil (its type (according to WRB), the region where it was taken from). The authors cite pH and OC values ​​for the assessment of soil buffering, but the content of particles <0.02 mm and <0.002 mm should be added.
2) In the light of the problems developed by the authors, the reader may not be clear why there was Soil for Uncultivated pots in the experiments? This simulates what kind of situation in reality?
3) If the authors indicated at the end of their Manuscript what logistic solutions can be offered for the delivery of wasted bread to the soils and the methods of their introduction, then there would be a certain completeness.
Conclusion. The last proposal is very important and one could add hypotheses for which soils (with what quality) the risks of such a dietary supplement would be risky or undesirable.
Specific Comments:
L 13 and 19 The introduction of an acronym like WB and its disclosure in the Abstract is not very good. Moreover, this is repeated on L 42. The reviewer believes that abbreviations on L 19 can be avoided.
Keywords: lactic acid bacteria fermentation; This phrase is cumbersome.
L 71 - 82 - compare %%. Authors put spaces and do not put them
L 79, 81 - 87 compare Authors put spaces and do not put them
L 71 extra point 70.82% .;
L 156 soil type and region?
L 133 and 162. The method is the same, but the links are different (19 and 20). Why?
Table 1. It is logical that the column with C: N was moved after TN
Table 1 and Table 2. The EC parameter is given with different units of measurement, is this justified?
Table 1 and Table 2. The TN parameter is given with different units of measurement. Is this the same Parameter? If not (total and mobile), you need to change the abbreviations.
References. Source 24 does not belong to scientific literature and is more appropriate in the text, at the first mention as a footnote.
L 550-551. This is "the tail from another cat" -
- Author 1, A.B.; Author 2, C.D.; Author 3, E.F. Title of Presentation. In Title of the Collected Work (if available), Proceedings of the Name of the Conference, Location of Conference, Country, Date of Conference; Editor 1, Editor 2, Eds. (if available); Publisher: City, Country, Year (if available); Abstract Number (optional), Pagination (optional).

Author Response
The article touches on an important topic in its research. The authors approached this problem using well-substantiated methodological experiments. The article is well structured and logical. There are, perhaps, no fundamental remarks, except for some suggestions for revision.
1) This reviewer, being a soil scientist, considers it necessary to indicate a specific soil (its type (according to WRB), the region where it was taken from). The authors cite pH and OC values ​​for the assessment of soil buffering, but the content of particles <0.02 mm and <0.002 mm should be added.
The classification of the soil and its particle size have been added to the text (153-157).
2) In the light of the problems developed by the authors, the reader may not be clear why there was Soil for Uncultivated pots in the experiments? This simulates what kind of situation in reality?
The use of uncultivated pots does not simulate any situation in reality. We tested soil with plants and without plant to to assess whether the changes observed were a consequence of the amendment addition or of the soil ecosystem interaction with the plant (rhizosphere effect). To better clarify this point, a new comment has been added in lines 158-161.
3) If the authors indicated at the end of their Manuscript what logistic solutions can be offered for the delivery of wasted bread to the soils and the methods of their introduction, then there would be a certain completeness.
As suggested, this aspect was better explored in lines 512-523.
Conclusion. The last proposal is very important and one could add hypotheses for which soils (with what quality) the risks of such a dietary supplement would be risky or undesirable.
Alkaline soils should be the best soil recipient for such residual biomasses since acidic biomasses can worsen the fertility of acidic soils since their application could enhance i) the release of Al, toxic for the root in high dose, from the mineral weathering; ii) the bioavailability of potentially toxic micronutrients and trace elements, such as Cr, Cu, etc., that in low doses are nutrients, while in higher dose are toxic.
A new comment was added to the revised text to clarify this point (500-504).
Specific Comments:
L 13 and 19 The introduction of an acronym like WB and its disclosure in the Abstract is not very good. Moreover, this is repeated on L 42. The reviewer believes that abbreviations on L 19 can be avoided.
As suggested, the acronyms were removed from the abstract.
Keywords: lactic acid bacteria fermentation; This phrase is cumbersome.
Ok. The keyword was modified.
L 71 - 82 - compare %%. Authors put spaces and do not put them
Ok. Spaces before % were aligned throughout the text.
L 79, 81 - 87 compare Authors put spaces and do not put them
Spaces before % were aligned throughout the text.
L 71 extra point 70.82% .;
The point was removed.
L 156 soil type and region?
As suggested new information was added (lines 153-157)
L 133 and 162. The method is the same, but the links are different (19 and 20). Why?
Both methods (Walkley-Black and the Springer-Klee based method by Ciavatta et al.) can be used for the determination of the OC content of soils, but only the latter can be applied to biomasses due to their higher OC content than soils that can saturate the K bichromate.
In fact, the Walkley-Black method is based on the use of 1 N K bichromate, while the Ciavatta includes the use of the 2 N K bichromate. Thus, it is not possible to use the Walkley-Black method to biomasses.
A comment has been added to the revised text to clarify this point (L 147).
Table 1. It is logical that the column with C: N was moved after TN
Ok. As suggested the C/N column was moved after TN.
Table 1 and Table 2. The EC parameter is given with different units of measurement, is this justified?
The parameter is the same, but the method used for the determination of soil EC is different from the one for the determination of biomasses, thus commonly associated to different units. Nevertheless, the units have been standardized in the revised Tables.
Table 1 and Table 2. The TN parameter is given with different units of measurement. Is this the same Parameter? If not (total and mobile), you need to change the abbreviations.
It is the same parameter. For biomasses, total nitrogen content is commonly expressed as percentage, while g kg-1 is commonly used in literature for soil and substrates.
References. Source 24 does not belong to scientific literature and is more appropriate in the text, at the first mention as a footnote.
According to Foods authors guidelines, websites should be placed in the reference list specifying the URL and the last time it was accessed.
L 550-551. This is "the tail from another cat" -
- Author 1, A.B.; Author 2, C.D.; Author 3, E.F. Title of Presentation. In Title of the Collected Work (if available), Proceedings of the Name of the Conference, Location of Conference, Country, Date of Conference; Editor 1, Editor 2, Eds. (if available); Publisher: City, Country, Year (if available); Abstract Number (optional), Pagination (optional).
It was an oversight; this part was removed.
Reviewer 2 Report
In the manuscript titled "Reuse of Wasted Bread as Soil Amendment " investigation regarding the potential of bread to be used as an organic soil amendment is presented.
The article is well prepared and I have only one hint to check line 296. I suppose it should be "decrease, from 63 to 59%".
Furthermore, the discussion/a few sentences regarding the economic perspectives of the procedure would be valuable. Is possible to implement this solution on an industrial scale.
Author Response
In the manuscript titled "Reuse of Wasted Bread as Soil Amendment " investigation regarding the potential of bread to be used as an organic soil amendment is presented.
The article is well prepared and I have only one hint to check line 296. I suppose it should be "decrease, from 63 to 59%".
The sentence was clarified (Line 340-342).
Furthermore, the discussion/a few sentences regarding the economic perspectives of the procedure would be valuable. Is possible to implement this solution on an industrial scale.
Implementing this solution to a larger scale is exactly what this study intended to foresee, however, to do that, more studies are necessary to fully exploit the potential of food waste as amendment. A more complete study of the overall amended soils ecosystem should be performed. And a better structured plan on how to manage the entire process is also essential, as well as collaborations with companies that produce fertilizers, to ensure a widespread supply.
To clarify these points, new comments concerning limitations and potential of the solution proposed, were added (Lines 512-523).
Reviewer 3 Report
Title
The title may be modified as “Reuse of wasted bread as soil amendment for enhanced biomass production in escarole (Cichorium endivia var. Cuartana)”
Abstract
There should be 1-2 sentence highlighting the importance of the conducted study
Introduction
A paragraph may be added on highlighting the importance of PGPR in improving growth and yield of crops.
The last section needs attention regarding highlighting the novelty with clear hypothesis and objectives
Tables
Under each table, there must be description of the values after mean values
Table 4 Correct this “Average n. of leaves/plant”
Results
It would be better to present the economic analysis of the tested amendment
Author Response
Title
The title may be modified as “Reuse of wasted bread as soil amendment for enhanced biomass production in escarole (Cichorium endivia var. Cuartana)”
The suggestion was appreciated, and the title changed according to it.
Abstract
There should be 1-2 sentence highlighting the importance of the conducted study
A sentence was added in lines 12-16
Introduction
A paragraph may be added on highlighting the importance of PGPR in improving growth and yield of crops.
The last section needs attention regarding highlighting the novelty with clear hypothesis and objectives
The introduction was updated as suggested (lines 63-66; 71-77)
Reviewer suggestion
Tables
Under each table, there must be description of the values after mean values
As suggested a sentence was added under each table.
Table 4 Correct this “Average n. of leaves/plant”
Done.
Results
It would be better to present the economic analysis of the tested amendment
I’m sorry, an accurate economic analysis of the whole production chain was not carried out in this step of the research. The evaluation of logistics and sustainability have been scheduled for the future, and open-field trials will follow. Nevertheless, the potential of the solution proposed, both by technical and economic point of views has been commented in this revised version of the manuscript (Lines 512-523).
Round 2
Reviewer 3 Report
The authors have incorporated the suggested changes.